# Predicted Absorption Performance of Cylindrical and Rectangular Permeable Membrane Space Sound Absorbers Using the Three-Dimensional Boundary Element Method

**Masahiro Toyoda [1,*], Kota Funahashi [2], Takeshi Okuzono [2] and Kimihiro Sakagami [2]**

[1] Department of Architecture, Faculty of Environmental and Urban Engineering, Kansai University, Yamate-cho, Suita, Osaka 564-8680, Japan

[2] Environmental Acoustics Lab., Department of Architecture, Grad. Sch. of Engineering, Kobe University, Rokko, Nada, Kobe 657-8501, Japan; kota_tako_214@yahoo.co.jp (K.F.); okuzono@port.kobe-u.ac.jp (T.O.); saka@kobe-u.ac.jp (K.S.)

[*] Correspondence: toyoda@kansai-u.ac.jp; Tel.: +81-6-6368-1924

**Abstract:** Three-dimensional, permeable membrane space sound absorbers have been proposed as practical and economical alternatives to three-dimensional, microperforated panel space sound absorbers. Previously, the sound absorption characteristics of a three-dimensional, permeable membrane space sound absorber were predicted using the two-dimensional boundary element method, but the prediction accuracy was impractical. Herein, a more accurate prediction method is proposed using the three-dimensional boundary element method. In the three-dimensional analysis, incident waves from the elevation angle direction and reflected waves from the floor are considered, using the mirror image. In addition, the dissipated energy ratio is calculated based on the sound absorption of a surface with a unit sound absorption power. To validate the three-dimensional numerical method, and to estimate the improvement in prediction accuracy, the results are compared with those of the measurements and two-dimensional analysis. For cylindrical and rectangular space sound absorbers, three-dimensional analysis provides a significantly improved prediction accuracy for any shape and membrane sample that is suitable for practical use.

**Keywords:** membrane; permeability; space sound absorber; boundary element method

## 1. Introduction

Problems due to insufficient sound absorption arise due to increased noise sources and increasing numbers of large-scale buildings. For example, insufficient sound absorption leads to excessive reverberation, lowering speech intelligibility. Consequently, the transmission performance of voice information deteriorates. During an emergency, this deterioration may threaten people's safety. To reduce and control the noise generated from various sources [1], and to create a quiet and comfortable sound environment, sound absorption is an important factor of building quality. Currently, natural fibers as absorptive materials are extensively investigated [2–6]. They are of recyclable and sustainable properties and show good absorption performance. Nevertheless, glass wool is still widely used because it is low cost and has high absorptivity.

Although glass wool is an economical sound absorbing material with a high sound absorption, it has some drawbacks [7]. First, it should be avoided in sanitary settings such as a hospital or precision machine factory because glass wool dust is harmful to humans. Second, durability issues have been reported in harsh environmental conditions of high temperature and humidity. Third, recycling

is challenging, due to its nonflammable nature. Fourth, recent studies have indicated that fibrous materials adversely affect the environment.

A perforated panel has also been widely used as an absorptive material. A perforated panel is a non-fibrous material and can overcome the drawbacks glass wool has. However, a perforated panel itself has a low sound absorption performance; it is generally utilized as a facing filled with a fibrous material. In this case, the sound absorption depends on the fibrous material inside the structure. Therefore, non-fibrous materials that can be used alone as sound absorbers are attracting much attention.

Recently, many sound absorbing materials with a porous structure, superior hygiene, and outstanding environmental characteristics have been developed [8]. One promising alternative is an economical and easy-to-process membrane similar to glass wool. Such membranes are mainly used in the walls and roofs of sports facilities and atriums, due to their unique properties (e.g., lightweight and flexible) [9]. Moreover, their translucency allows natural light to be incorporated into a building, realizing various light effects. They have high durability and weather resistance. Unlike glass wool, these membranes do not generate dust and are safe for humans and the environment. They also allow for free shaping facilitating operations, such as processing, installation, and transportation.

Previous studies have shown that a non-permeable membrane alone has almost no sound absorption [10]. Since they reflect sound perfectly at high frequencies, they are often used as stage reflectors in multipurpose halls [9]. On the other hand, in terms of permeable membranes, the sound absorption at low frequencies is higher as the surface density increases. At middle and high frequencies, the absorption coefficient converges around a 0.5 maximum with the appropriate flow resistance [11]. As long as a single-leaf membrane has no permeability, sufficient sound absorption performance cannot be obtained by itself. Additionally, vibration-type absorption does not occur if it lacks a backing absorption element [12].

Since it is difficult to obtain sufficient sound absorption performance by a membrane alone, numerous theoretical and experimental studies on the acoustic characteristics of membrane composite structures have been conducted. Studies on the acoustic properties of double-leaf membrane structures with impermeable or permeable membranes [13,14], especially permeable ones, have been reported as high-performance sound absorbers [15]. Consequently, the use of membranes as architectural materials should progress beyond walls and ceilings. For example, using membranes as an interior sound absorber may realize lightweight, highly designed, and maintenance-free sound absorbers.

On the other hand, microperforated panels (MPPs) [16], which are improved perforated plates, have been developed as substitutes for fibrous materials. An MPP is a thin plate or film less than 1-mm thick, where a large number of small holes that are 1-mm in diameter or less are drilled at a perforation ratio of about 1%. An MPP has excellent durability and weather resistance. It is suitable for high temperature and high humidity environments. Moreover, dust from fibrous materials is not an issue. Since an MPP can be manufactured using any material, it is attractive in terms of designability. Depending on the material, it can be recycled, reducing the disposal cost. Typically, an MPP is installed in front of a rigid wall with an in-between air layer. Sound absorption occurs based on the Helmholtz-type resonator formed by small holes and the air layer. The absorption characteristics of MPP absorbers have been predicted analytically using the equivalent electrical circuits (for example, see [16]) or physical consideration of small perforations (for example, see [17]). In addition, the practical absorption performance in a relatively complex situation has been recently estimated by numerical methods like the finite element method (FEM) [18–22], the boundary element method (BEM) [23], the finite-difference time-domain (FDTD) method [24], and the computational fluid dynamics (CFD) [25,26].

If an MPP as a space sound absorber does not have a back wall, sound absorption via a mechanism other than the Helmholtz resonator may occur, due to its air permeability similar to a membrane. Multiple configurations have been proposed as a space sound absorber using MPPs: two MPPs arranged in parallel, called double-leaf MPPs (DLMPPs) [27,28]; three MPPs arranged in the same way,

called triple-leaf MPPs (TLMPPs) [29]; and an MPP combined with a membrane [30]. Although these are effective space sound absorbers, they are flat. Such configurations can be used as partitions, but cannot be installed in non-flat structures. Three-dimensional MPP space sound absorbers have been proposed as an MPP space sound absorber with a wide range of applications and more handling convenience. Two examples are a cylindrical MPP space sound absorber (CMSA) and a rectangular MPP space sound absorber (RMSA). CMSA is formed by rolling one sheet of an MPP, while RMSA is formed using four flat MPPs. Experimental studies have shown that these have significant sound absorption performances [31–33]. Furthermore, a numerical prediction method based on the two-dimensional boundary element method has been proposed and validated by a comparison with experimental results [23].

An MPP is expensive, and widespread use remains a challenge. From this viewpoint, less expensive, easy-to-manufacture, three-dimensional permeable membranes have been proposed as more practical alternatives [34]. Experiments have demonstrated that three-dimensional permeable membrane space sound absorbers have sufficient sound-absorbing performance. Moreover, a numerical prediction method for their absorption performance based on the two-dimensional boundary element method has been proposed [34]. Unlike the case of three-dimensional MPP space sound absorbers, the two-dimensional numerical method does not reproduce the absorption performance of permeable membrane space sound absorbers; it predicts peaks and dips in the absorption coefficient of a heavy permeable membrane with a high flow resistance, which are not observed in the measurement data. Hence, an accurate prediction method for permeable membrane space sound absorbers is necessary.

In this study, a numerical prediction method based on the three-dimensional boundary element method is proposed for the sound absorption performance of a permeable membrane space sound absorber. Permeable membranes made of polyethylene terephthalate (PET)- and polytetrafluoroethylene (PTFE)-coated glass fiber woven fabrics are herein investigated. These membranes have high durability and recyclability [7,35]. Particularly, PET is a completely recyclable material [7]. These membranes therefore show sustainable properties. For cylindrical and rectangular permeable membrane absorbers, the energy dissipation ratios calculated by two- and three-dimensional boundary element methods are compared with the measured ones obtained by the reverberation room method, to show that the three-dimensional numerical method is a more accurate prediction tool.

## 2. Formulation

The three-dimensional boundary element method is formulated for a cylindrical, permeable membrane space sound absorber (CPMSA) and a rectangular, permeable membrane space sound absorber (RPMSA). The small tension of the permeable membrane has a negligible influence on the acoustic characteristics [10]. Here, the tension of the membrane is ignored.

### 2.1. Model

The CPMSA and RPMSA, which are three-dimensional sound absorbers discussed experimentally in a previous study [34], are examined theoretically. Similar to the previous study, two different sizes of CPMSA and RPMSA (e.g., four samples) were considered. CPMSA and RPMSA were made by loosely stretching permeable membranes on the sides of rigid thin frames, where the upper ends were opened. The sound absorbers were installed on a rigid floor in a reverberation room. Figure 1 schematically depicts the measurements of CPMSA and RPMSA. Theoretical considerations are performed on a CPMSA and RPMSA placed in a free space under a plane wave incidence at an elevation angle $\theta$ and an azimuth angle $\gamma$.

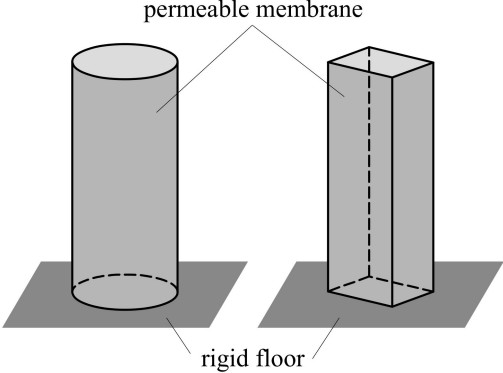

**Figure 1.** Schematic diagram of the cylindrical, permeable membrane space sound absorber (CPMSA) and rectangular, permeable membrane space sound absorber (RPMSA) of the experimental setup.

In the actual measurement, the energy incident on the sound absorbers includes not only the direct incident, but also indirect ones after reflection on the floor surface. However, in the theoretical analyses, if the calculation model is made with the same dimensions as the real sample, the indirect incident energy reflected on the floor surface cannot be considered, because the sound absorbers are placed in free space. In this study, doubling the length of CPMSA and RPMSA in the long axis direction considers the mirror images and the indirect incident energy.

Figure 2 shows the calculation models of CPMSA and RPMSA under a unit amplitude plane wave incidence. Considering the symmetry, the azimuth angle is set to 0 degrees for the CPMSA and 0 to 45 degrees for the RPMSA. It has been theoretically proven that the sound absorption coefficient obtained by the reverberation room method corresponds to the energy dissipation ratio in the diffuse sound field [36,37]. Previous studies have shown that the measured values for the sound absorption coefficient of the reverberation room method for space sound absorbers agree well with the calculated values for the sound field incidence of a plane wave [36].

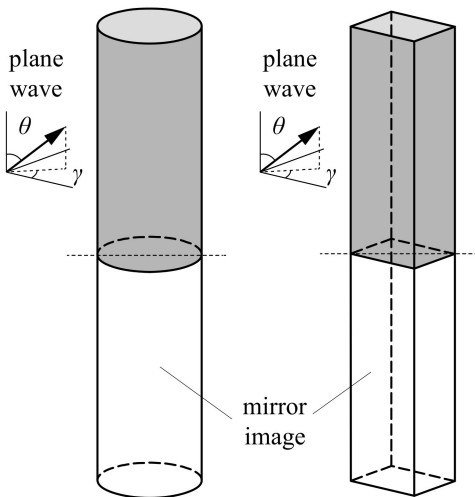

**Figure 2.** Schematic diagrams of the CPMSA and RPMSA in the theoretical consideration.

*2.2. Boundary Integral Equation*

Figure 3 shows the three-dimensional model for deriving the boundary integral equation. Region $\Omega_1$ is bounded externally by a sphere $\Sigma$, and internally by $\Gamma$ and small spheres $\sigma_s$ and $\sigma_p$. Region $\Omega_2$ is bounded externally by $\Gamma$; p is the sound receiver, which is located at the center of $\Sigma$. In addition, $\sigma_s$ and $\sigma_p$ have centers s and p, respectively, and small radii $\varepsilon$. Variable s is the sound source, q is a point located on the boundary, and **n** is the inward normal. $\Gamma$ has transfer admittance ratio *A*. The time

factor $\exp(-i\omega t)$ is suppressed throughout, where i is an imaginary unit, $\omega$ is the angular frequency, and $t$ is time. Considering the Green's identity for region $\Omega_1$ yields

$$\int_{\Omega_1} \left( f\nabla^2 g - g\nabla^2 f \right) dV = \int_{\Sigma+\sigma_s+\sigma_p+\Gamma} \left( f\frac{\partial g}{\partial n} - \frac{\partial f}{\partial n}g \right) dS, \tag{1}$$

where $f$ and $g$ are continuous and smooth functions, respectively. Velocity potential $\Phi$ and basic solution $G$ are now substituted into $f$ and $g$ in Equation (1), respectively. Assuming $\Phi$ and $G$ satisfy the three-dimensional Helmholtz equation, the boundary integral equation can be derived for region $\Omega_1$. $G$ can be given by

$$G\left(\mathbf{r}_p, \mathbf{r}_q\right) = \frac{e^{ikr_{pq}}}{4\pi r_{pq}}, \tag{2}$$

where r is a position vector, $k$ is the wavenumber, and $r_{pq}$ is the distance between p and q. The integral over $\sigma_s$ in Equation (1) can be expressed by taking the limit $\varepsilon$ to 0 as

$$\int_{\sigma_s} \left\{ \Phi\{\mathbf{r}_q\}\frac{\partial G\{\mathbf{r}_p, \mathbf{r}_q\}}{\partial \mathbf{n}_q} - \frac{\partial \Phi\{\mathbf{r}_q\}}{\partial \mathbf{n}_q}G\{\mathbf{r}_p, \mathbf{r}_q\} \right\} dS = \varphi_d\left(\mathbf{r}_p\right), \tag{3}$$

where $\varphi_d$ is the velocity potential given by the direct sound. Similar to the derivation of Equation (3), the integral over $\sigma_p$ in Equation (1) can be obtained as

$$\int_{\sigma_p} \left\{ \Phi\{r_q\}\frac{\partial G\{\mathbf{r}_p, \mathbf{r}_q\}}{\partial \mathbf{n}_q} - \frac{\partial \Phi\{\mathbf{r}_q\}}{\partial \mathbf{n}_q}G\{\mathbf{r}_p, \mathbf{r}_q\} \right\} dS = -C\left(\mathbf{r}_p\right)\varphi\left(\mathbf{r}_p\right), \tag{4}$$

where $C(\mathbf{r}_p)$ is the ratio of the included-part solid angle of $\sigma_p$ in $\Omega_1$ to $4\pi$. When p is located on a smooth boundary, $C(\mathbf{r}_p)$ becomes 1/2. If the radius of $\Sigma$ is infinite, the integral over $\Sigma$ in Equation (1) can be neglected, because the Sommerfeld radiation condition [38] is given by

$$\left|r_{pq}\Phi\left|\mathbf{r}_q\right|\right| < K, \quad \sqrt{r_{pq}}\left\{ \frac{\partial \Phi\{r_q\}}{\partial r_{pq}} - ik\Phi\{r_q\} \right\} \to 0 \quad \left(r_{pq} \to \infty\right), \tag{5}$$

where $K$ is a finite real number. After all, by substitution of Equations (3) and (4) into Equation (1), one can obtain the boundary integral equation for region $\Omega_1$ as

$$\varphi_d\left(\mathbf{r}_p\right) + \int_{\Gamma} \left\{ \varphi_1\{\mathbf{r}_q\}\frac{\partial G\{\mathbf{r}_p, \mathbf{r}_q\}}{\partial \mathbf{n}_q} - \frac{\partial \varphi_1\{\mathbf{r}_q\}}{\partial \mathbf{n}_q}G\{\mathbf{r}_p, \mathbf{r}_q\} \right\} dS = \tfrac{1}{2}\varphi_1\left(\mathbf{r}_p\right) \quad (p \in \Gamma), \tag{6}$$

where $\varphi_1$ is the velocity potential on $\Gamma$ of the $\Omega_1$ side. Considering a similar approach, that for region $\Omega_2$ can be obtained as

$$-\int_{\Gamma} \left\{ \varphi_2\{\mathbf{r}_q\}\frac{\partial G\{\mathbf{r}_p, \mathbf{r}_q\}}{\partial \mathbf{n}_q} - \frac{\partial \varphi_2\{\mathbf{r}_q\}}{\partial \mathbf{n}_q}G\{\mathbf{r}_p, \mathbf{r}_q\} \right\} dS = \tfrac{1}{2}\varphi_2\left(\mathbf{r}_p\right) \quad (p \in \Gamma), \tag{7}$$

where $\varphi_2$ is the velocity potential on $\Gamma$ of the $\Omega_2$ side. Assuming $v(\mathbf{r}_q)$, the particle velocity at q, yields

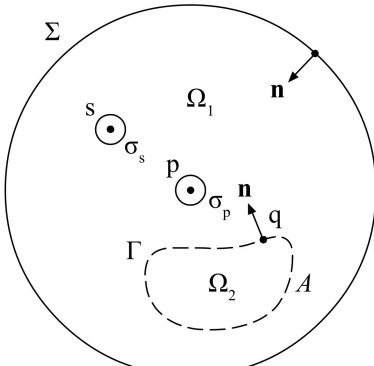

**Figure 3.** Three-dimensional model to derive the boundary integral equation.

$$\frac{\partial \varphi_1(\mathbf{r_q})}{\partial \mathbf{n_q}} = \frac{\partial \varphi_2(\mathbf{r_q})}{\partial \mathbf{n_q}} = -v(\mathbf{r_q}) = -ikA(\mathbf{r_q})\{\varphi_1\{\mathbf{r_q}\} - \varphi_2\{\mathbf{r_q}\}\}. \tag{8}$$

Differentiating the equation obtained by adding Equations (6) and (7) in terms of the normal for p and considering Equation (8), one can obtain

$$\frac{\partial \varphi_d(\mathbf{r_p})}{\partial \mathbf{n_p}} + \int_\Gamma \left\{ \varphi\{r_q\} \frac{\partial^2 G\{r_p, r_q\}}{\partial n_p \partial n_q} \right\} dS = -ikA(\mathbf{r_p})\varphi(\mathbf{r_p}) \quad (\text{p} \in \Gamma) \ , \tag{9}$$

where $\varphi = \varphi_1 - \varphi_2$. Discretizing $\Gamma$ with $N$ constant elements, an approximate algebraic equation of Equation (9) can be obtained as

$$\frac{\partial \varphi_{d,i}}{\partial \mathbf{n}_i} + \sum_{j=1}^{N} \varphi_j \int_{\Gamma_j} \left\{ \frac{\partial^2 G\{\mathbf{r}_i, \mathbf{r_q}\}}{\partial \mathbf{n}_i \partial \mathbf{n_q}} \right\} dS = -ikA_i \varphi_i \quad (i = 1, \cdots, N) \ , \tag{10}$$

where

$$\frac{\partial \varphi_{d,i}}{\partial \mathbf{n}_i} = \frac{\partial e^{ik\{\mathbf{h}\{\theta, \gamma\} \cdot \mathbf{r}_i\}}}{\partial \mathbf{n}_i} = ik\{\mathbf{h}\{\theta, \gamma\} \cdot \mathbf{n}_i\}e^{ik\{\mathbf{h}\{\theta, \gamma\} \cdot \mathbf{r}_i\}}, \tag{11}$$

$$\frac{\partial^2 G(\mathbf{r}_i, \mathbf{r_q})}{\partial \mathbf{n}_i \partial \mathbf{n_q}} = \frac{e^{ikr_{P_iq}}}{4\pi r_{P_iq}^3} \left[ \left[1 - ikr_{P_iq}\right]\cos[\mathbf{n}_i, \mathbf{n_q}] + \left\{3\{ikr_{P_iq} - 1\} + k^2 r_{P_iq}^2\right\}\cos[\mathbf{r}_{P_iq}, \mathbf{n}_i]\cos[\mathbf{r}_{P_iq}, \mathbf{n_q}]\right], \tag{12}$$

In Equation (11), $\mathrm{h}(\theta, \gamma) = (\cos\gamma\sin\theta, \sin\gamma\sin\theta, \cos\theta)$ is a unit vector, which means the direction of the plane wave. In Equation (12), $\mathrm{p}_i$ means the center of $i$th element. For calculation of the integral in Equation (12), the fourth-order Gauss–Legendre quadrature is adopted for $i \neq j$; on the other hand, the integral that has hyper singularity for $i = j$ is calculated using Terai's method [39]. By solving the simultaneous equations of Equation (10), one can obtain the potential differences $\varphi_i$.

## 2.3. Permeable Membrane Admittance

Figure 4 schematically diagrams a permeable membrane under pressure difference $p_1 - p_2$, where $p_1$ and $p_2$ are the sound pressures on the incident and transmission sides of the membrane, respectively. Here, $v_b$ is the velocity of the membrane vibration, and $v_a$ is the air particle velocity transmitting through the membrane, which is given as a relative value to $v_b$; $v$ is the averaged particle velocity of air near the membrane. Considering the assumptions for Figure 4, the continuity of volume velocity can be expressed as

$$v = \zeta v_a + v_b, \tag{13}$$

where $\zeta$ is the surface porosity of the membrane. The equation of motion for the membrane can be expressed as

$$m\frac{\partial v_b}{\partial t} = p_1 - p_2,\tag{14}$$

where $m$ is the surface density of the membrane. The definition of flow resistance $r$ is given by

$$r = \frac{p_1 - p_2}{v},\tag{15}$$

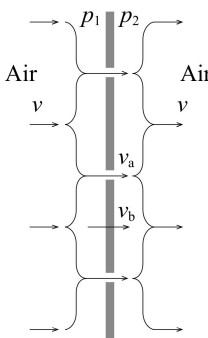

**Figure 4.** Schematic diagram of a permeable membrane under a sound pressure difference.

Now, note that $v$ in Equation (15) is commonly measured under a static air flow. In that case, $v_b = 0$, and $v$ in Equation (15) becomes $\zeta v_a$. Therefore, Equation (15) can be rewritten as

$$r = \frac{p_1 - p_2}{\zeta v_a},\tag{16}$$

Considering Equations (13), (14), and (16), the total transfer impedance of membrane $z$ is obtained as [40]

$$z = \frac{p_1 - p_2}{v} = \left(-\frac{1}{i\omega m} + \frac{1}{r}\right)^{-1},\tag{17}$$

Therefore, the transfer admittance ratio $A_i$ in Equation (10) is given by

$$A_i = \rho c\frac{1}{z} = \rho c\left(-\frac{1}{i\omega m} + \frac{1}{r}\right),\tag{18}$$

where $\rho$ and $c$ are the density and sound speed of air, respectively.

*2.4. Dissipated Energy Ratio*

Sound absorption performance of CPMSAs and RPMSAs should be evaluated by the dissipated energy ratio instead of sound absorption coefficient, which is calculated only from the reflection coefficient, because sound can be transmitted through the structures. The dissipated energy ratio can be obtained as the difference of the absorption and transmission coefficients, $\alpha - \tau$, which can be calculated from the potential differences $\varphi_i$. The dissipated energy in the $j$th element $W_j(\theta, \gamma)$ under a plane wave of incident angles $\theta$ and $\gamma$ can be given by

$$W_j(\theta, \gamma) = \frac{1}{2}\mathrm{Re}\{p_j \cdot \{-v_j\}^*\}\Delta S_j,\tag{19}$$

where the asterisk denotes the complex conjugate, $\Delta S_j$ indicates the area of the $j$th element, and $p_j$ is the sound pressure difference between the incident and transmitted sides of the membrane, which can be obtained as

$$p_j = \rho \frac{\partial \varphi_j}{\partial t} = -i\rho\omega\varphi_j, \tag{20}$$

where $v_j$ is the particle velocity on the membrane surface, which can be calculated by

$$v_j = -ikA_j\varphi_j. \tag{21}$$

Then, the total dissipated energy $W_a(\theta, \gamma)$ can be written as

$$W_a(\theta, \gamma) = \sum_{j=1}^{N} W_j(\theta, \gamma) = \frac{(\rho\omega)^2}{2\rho c} \sum_{j=1}^{N} Re\{A_j\}|\varphi_j|^2 \Delta S_j, \tag{22}$$

In the analysis via the two-dimensional model [34], the ratio of the sum of the energy consumed by each element to the incident energy on the apparent surface area of the sound absorber under the oblique incidence is assumed to correspond to the dissipated energy ratio. However, estimating the apparent surface area of the sound absorber under oblique incidence is difficult in three-dimensional models. For example, the energy into the upper-end opening of the sound absorber cannot be considered adequately. This study sets the criteria as the energy consumed by the plane with a unit absorption power under oblique incidence of the plane wave. The absorption power of the sound absorber is estimated by the ratio of the sum of the energy dissipated by each element to the criteria, while $\alpha - \tau$ is calculated by dividing the power by the surface area of the membrane.

Consider a plane with a unit absorption power under a unit amplitude plane wave incidence. The dissipated energy of the plane $W_i(\theta, \gamma)$ can be given by

$$W_i(\theta, \gamma) = \frac{(\rho\omega)^2}{2\rho c} \cos\theta, \tag{23}$$

Consequently, considering the symmetric property, the dissipated energy ratio of the CPMSA can be expressed as

$$\alpha - \tau = \frac{1}{S} \frac{\int_0^{80} W_a(\theta, 0) \sin\theta d\theta}{\int_0^{80} W_i(\theta, 0) \sin\theta d\theta}, \tag{24}$$

where $S$ denotes the total area of membrane. The energy ratio of RPMSA can be written as

$$\alpha - \tau = \frac{1}{S} \frac{\int_0^{80} \int_0^{45} W_a(\theta, \gamma) \sin\theta d\gamma d\theta}{\int_0^{80} \int_0^{45} W_i(\theta, \gamma) \sin\theta d\gamma d\theta}, \tag{25}$$

## 3. Results and Discussion

Table 1 shows the physical properties of the membranes used in the measurements [34]. The pictures of the experimental setup are shown in Figure 5. Two types of samples with a height of 1 m were prepared: 1 m$^2$ and 2 m$^2$ membranes. They were discretized into squares of 0.025 m. Considering the mirror image, the total number of elements was 3200 for the 1 m$^2$ membrane and 6400 for the 2 m$^2$ one.

**Table 1.** Physical properties of the membranes.

| Membrane | Surface Density [kg/m$^2$] | Flow Resistance [Pa s/m] |
|----------|----------------------------|--------------------------|
| A | 0.065 | 196 |
| B | 0.120 | 462 |
| C | 0.495 | 1087 |

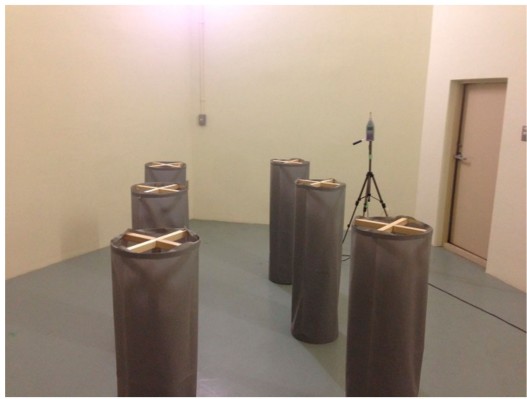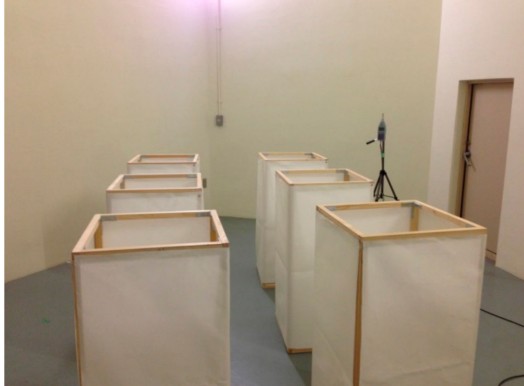

**Figure 5.** Pictures of the experimental setups. The left figure includes 1 m$^2$ CPMSA made of membrane A, and the right figure includes 2 m$^2$ RPMSA made of membrane C.

### 3.1. Cylindrical, Permeable Membrane Space Sound Absorber

Figure 6 compares the results from two-dimensional analysis, three-dimensional analysis, and measurements for the cylindrical samples with membranes A, B, and C of $1/\pi$ m and $2/\pi$ m diameters. For example, B2 indicates the case with membrane B with a $2/\pi$ m diameter. The values of the measurement and the two-dimensional analysis are from [34]. The horizontal axis indicates the center frequency of the 1/3-octave band, while the vertical axis indicates the dissipated energy ratio in the analysis and the measured sound absorption coefficient obtained by the reverberation room method.

The calculation for the dissipated energy ratio varied in the analysis. In the two-dimensional analysis, the dissipated energy ratio was taken as the sum of energy consumed in each element to the incident energy on the apparent surface area of the sound absorber, divided by the surface area of the membrane. As shown in Figure 6A1,A2, in cases with a very light membrane A, the absorption characteristics obtained by the two-dimensional analysis can express the general tendency of the measurement. Since the differences between the predicted and measured data are within about 0.1, the prediction accuracy of the two-dimensional analysis is sufficient for practical use. However, the absorption characteristics obtained by the three-dimensional analysis better reproduce the measurement and have a higher prediction accuracy than those of the two-dimensional analysis, because the three-dimensional analysis considers the mirror image of the sound absorber in order to take the incident wave reflected by the floor into account. In addition, the dissipated energy ratio was calculated based on the energy consumed by the plane with a unit of absorption power. These assumptions between the two- and three-dimensional analyses must be the factor for the differences in quantitative prediction accuracy. For $1/\pi$ m, the prediction accuracy is comparable to that of two-dimensional analysis from 2 to 4 kHz. However, the three-dimensional prediction is on the safe side compared to the two-dimensional one. Consequently, even in the $1/\pi$ m case, three-dimensional analysis should be employed. The discrepancies observed in the low frequency range, below 125 Hz, are attributed to the lack of diffusivity of the small volume reverberation chamber used in the measurement. The volume of the reverberation chamber used in the experiment is 130 m$^3$, which is slightly smaller than the criteria 150 m$^3$ defined in JIS A 1409:1998 (ISO 354:1985 compatible). Therefore, the measured absorption coefficients at low frequencies seem to be less reliable.

Similar to the case with very light membrane A, the prediction accuracy of the sound absorption coefficient by the three-dimensional analysis significantly improves compared to the two-dimensional one for membrane B (Figure 6B1,B2). At high frequencies, an error larger than 0.1 in the two-dimensional results falls within about 0.02 in the three-dimensional ones. Furthermore, a sharp dip occurs at 630 Hz in the two-dimensional analysis of Figure 6B1, but not in in the three-dimensional analysis or the measurement. In [34], it is inferred that the cause of this sharp dip is because three-dimensional phenomena are not considered in the two-dimensional analysis. Examples of such phenomena include

the sound field inside the sound absorber being disturbed by the oblique incident component in the elevation direction, which alleviates the horizontal standing wave generated inside the sound absorber and the modes in the vertical direction. Consequently, the two-dimensional analysis does not correspond well with the measurement. On the other hand, three-dimensional analysis can reproduce such phenomena accurately, mitigating the dip.

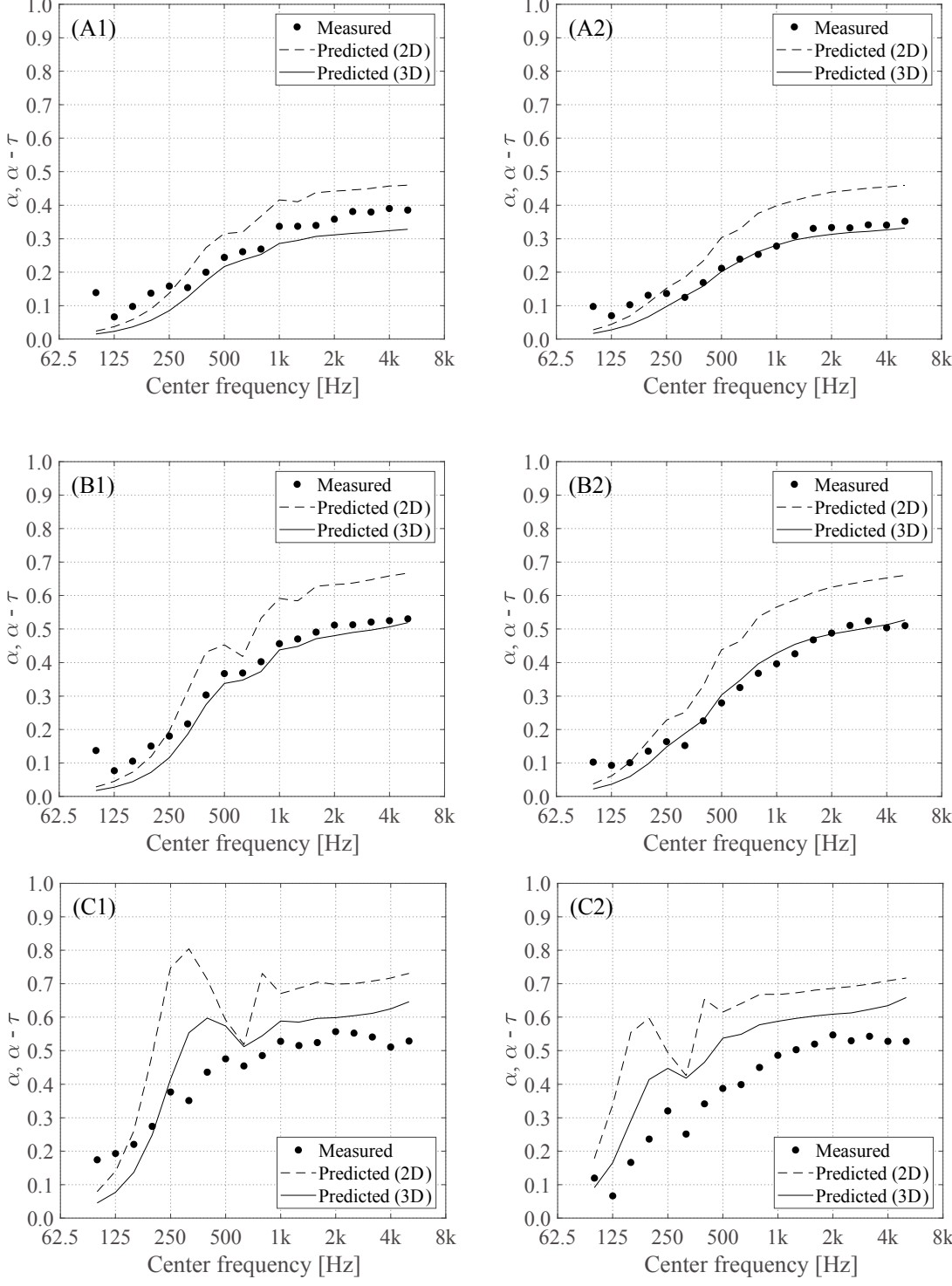

**Figure 6.** Comparisons between two-dimensional analysis, three-dimensional analysis, and measurements for CPMSAs. For example, (B2) indicates the results for membrane B with a $2/\pi$ m diameter.

In the case with the relatively heavy membrane C, neither the two- nor three-dimensional analyses reproduce the measurement well enough for practical use (Figure 6C1,C2). Due to the high flow resistance of the membrane, relatively strong resonances should occur inside. The two-dimensional analysis overestimates this resonance, and predicts a steep peak and dip. Although the peak and dip relax slightly in the three-dimensional analysis, it does not reproduce the measurement. These discrepancies may be due to inaccurate manufacturing of the cylindrical specimens in the measurement. Therefore, the measurement could have an azimuth direction dependence. Regardless, the three-dimensional analysis has a much better prediction accuracy than the two-dimensional analysis.

Although a previous study suggested that two-dimensional analysis for the sound absorption coefficients of CMSAs with the same dimensions has sufficient prediction accuracy [23], the prediction accuracy of CPMSAs is not predicted well with two-dimensional analysis. This is due to the difference in the sound absorbing mechanism between an MPP and a permeable membrane. A permeable membrane, which is a porous sound-absorbing material, mainly absorbs sound through airflow resistance, and the sound absorption coefficient is determined by the particle velocity on the membrane surface. Consequently, the sound field inside the absorber significantly affects the sound absorption characteristics. On the other hand, an MPP mainly absorbs sound by resonance. That is, the air in the hole of an MPP vibrates with the back air volume like a spring in order to generate sound absorption. Therefore, the influence of the sound field itself inside the absorber is weak. This difference in the sound absorbing mechanism between an MPP and a permeable membrane is responsible for the insufficient prediction accuracy by two-dimensional analysis for CPMSAs.

The required computational times were 1.18 s for the two-dimensional model and 345.48 s for the three-dimensional model, using dual Intel Xeon processors E5-2687W 3.1 GHz (20 cores in total). The increase of time is due to the increase of discretized elements and the integral for the elevation angle. In addition, the size of the CPMSA is larger, and the calculation cost increase exponentially. Although the time consumption is much higher for the three-dimensional case, the prediction accuracy is better, as shown above. In particular when the height-perimeter ratio is low, the difference from the two-dimensional model, where the height–perimeter ratio is infinite, becomes large, and the accuracy of two-dimensional analysis deteriorates significantly [23]; on the other hand, considering the results of three-dimensional analyses for 1 m$^2$ and 2 m$^2$ cases, the prediction accuracy remains, regardless the height–perimeter ratio.

### 3.2. Rectangular, Permeable Membrane Space Sound Absorber

Figure 7 compares the two-dimensional analysis, three-dimensional analysis, and measurements for the rectangular samples with membranes A, B, and C, with 0.25 m and 0.5 m sides. The general tendency is the same as that of the cylindrical type. For all membranes, three-dimensional analysis provides a better prediction accuracy than two-dimensional analysis. However, the correspondence with the measurement is slightly higher for the rectangular type. Because it is easier to more accurately manufacture rectangular specimens than cylindrical ones, the dependence on the azimuth direction can be precisely considered, resulting in an improved prediction accuracy.

The required computational times were 6.85 s for the two-dimensional model and 2832.05 s for the three-dimensional one. Comparing the time consumptions with those of CPMSAs, the required time for RPMSAs increases, because RPMSAs need an additional integral for the azimuth angle. However, similarly to CPMSAs, the prediction accuracy remains, regardless the height–perimeter ratio.

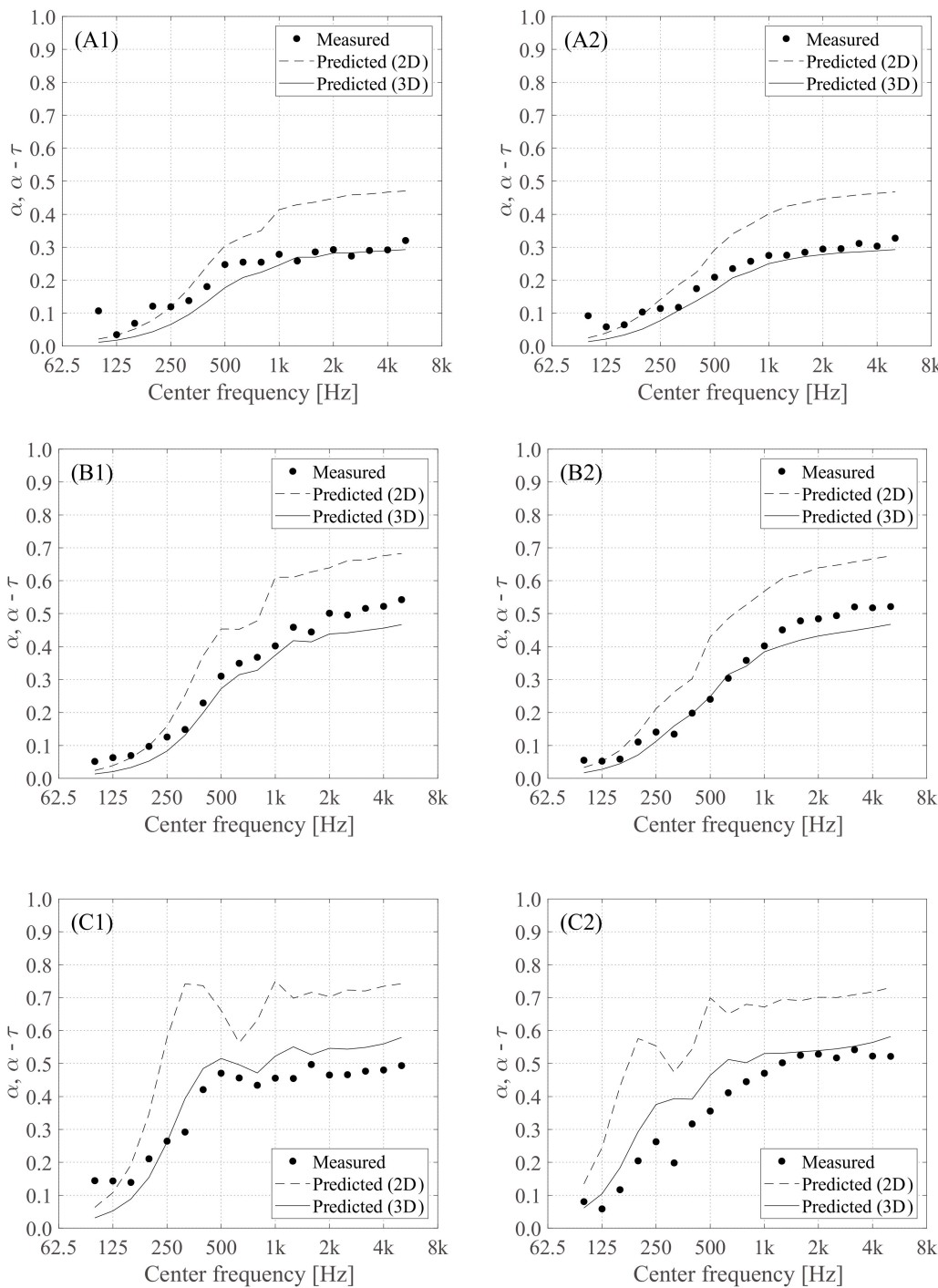

**Figure 7.** Comparisons between two-dimensional analysis, three-dimensional analysis, and measurements for RPMSAs. For example, (B2) indicates the results for membrane B with a $2/\pi$ m diameter.

### 3.3. Comparison with Glass Wool

Figure 8 shows the comparison between the predicted absorption characteristics of 1 m² CPMSAs and RPMSAs made of membranes A, B, and C, and two types of glass wool space sound absorbers that are the same sizes as the CPMSA and RPMSA. The two types of density and flow resistivity of glass wool are assumed to be 32 kg/m³ and 80 kg/m³ and 10,000 and 30,000 Pa s/m², respectively. The thickness is assumed to be 10 mm. Because the thickness is sufficiently small, the absorption performance can be predicted by the methods proposed in this study. According to the assumptions

above, the surface density and flow resistance are set to 0.32 kg/m² and 0.80 kg/m² and 100 Pa s/m² and 300 Pa s/m², respectively. The flow resistance is lower than that of membranes with the same surface density. Therefore, the results of absorption performance of the glass wool space sound absorbers are relatively flat over all frequencies. Higher sound absorption performance can be obtained at low frequencies by relatively larger surface densities [41]; on the other hand, high absorption cannot occur at high frequencies, because of the membranes' high permeability. CPMSAs and RPMSAs show superior absorption performance to glass wool space sound absorbers at high frequencies, and these comparisons illustrate the benefit of CPMSAs and RPMSAs.

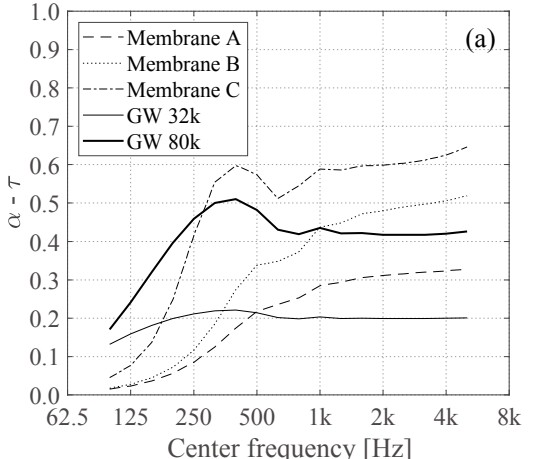 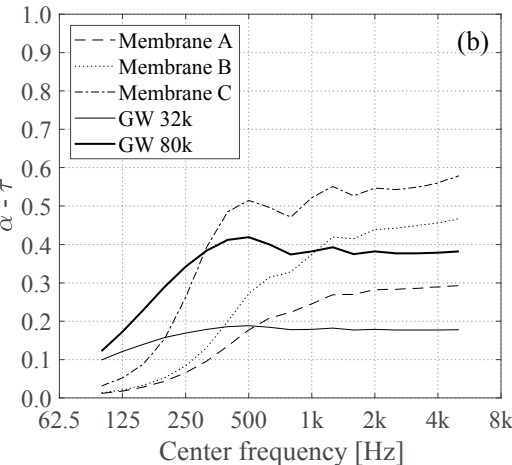

**Figure 8.** Comparison between (a) 1 m² CPMSAs and (b) 1 m² RPMSAs made of membranes A, B, and C and glass wool space sound absorbers that are the same sizes as the CPMSA and RPMSA.

## 4. Conclusions

We propose a numerical prediction method using the three-dimensional boundary element method for the sound absorption performance of CPMSA and RPMSA. The energy dissipation ratios are evaluated, dividing the ratios of the sum of the energy consumed by the element to the energy consumed by a plane, with a unit of absorption power by the surface area of the membrane. The energy dissipation ratios calculated by the two- and three-dimensional boundary element methods are compared with the measured absorption coefficients obtained by the reverberation room method. The three-dimensional numerical method provides a better prediction accuracy than the two-dimensional one, because the three-dimensional analysis can consider phenomena that the two-dimensional one cannot. Although the absorption performance of CMSA and RMSA can be predicted with practical and sufficient accuracy by two-dimensional analysis, three-dimensional analysis is necessary to predict the absorption performance of CPMSA and RPMSA, due to the different absorption mechanisms between MPPs and permeable membranes.

**Author Contributions:** M.T. served as the methodology adviser and wrote the original draft. K.F. oversaw the project and theoretical analysis. T.O. conducted data analysis and helped draft the manuscript. K.S. supervised data analysis and helped draft the manuscript.

**Funding:** This research received no external funding.

**Conflicts of Interest:** The authors declare no conflict of interest.

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
