# Peer review of "Predicted Absorption Performance of Cylindrical and Rectangular Permeable Membrane Space Sound Absorbers Using the Three-Dimensional Boundary Element Method"

_sustainability, doi:10.3390/su11092714_

Round 1

Reviewer 1 Report

Predicted absorption performance of cylindrical and rectangular permeable membrane space sound absorbers using the three-dimensional boundary element method

ID: 493744

The general comments of this reviewer are the following:

Line 35: “Currently, sound absorbing materials are commonly composed of fibrous materials such as glass wool”

Changing this sentence is recommended. In the last decade, there are many authors in the search of glass wool alternatives as sound absorbing fibrous material. Some examples are cited below:

E. Taban, A. Khavanin, M. Faridan, S.E. Samaei, K. Samimi, R. Rashidi. Comparison of acoustic absorption characteristics of coir and date palm fibers: experimental and analytical study of green composites. International Journal of Environmental Science and Technology. March 2019.

Berardi U, Iannace G (2015) Acoustic characterization of naturals fibers for sound absorption applications. Build Environ 94: 840-852.

Quintana A., Alba J., del Rey R., Guillen I. Comparative Life Cycle Assessment of gypsum plasterboard and a new kind of bio-based epoxy composite containing different natural fibers. Journal of Cleaner Production, 185, 408-420. 2018.

del Rey R., Uris A. Alba J., Candelas P. Characterization of Sheep Wool as a Sustainable Material for Acoustic Applications. Materials, 10, 11. 2017.

From line 37 to 42: Although glass wool is an economical sound absorbing material with a high sound absorption, it has some drawbacks. First, it should be avoided in sanitary settings such as a hospital or precision machine factory because glass wool dust is harmful to humans. Second, durability issues have been reported in harsh environmental conditions of high temperature and humidity. Third, recycling is challenging due to its nonflammable nature. Fourth, recent studies have indicated that fibrous materials adversely affect the environment.”

I absolutely agree with this paragraph but it is convenient to complete it with some references. This paragraph justify the publication of the article in Sustainability journal.

From line 43  to line 46: “Considering these drawbacks, non-fibrous materials are attracting much attention as sound absorbing materials. One example is a perforated panel. Because a perforated panel itself has a low sound absorption performance, it is generally utilized as a facing filled with a fibrous material. In this case, the sound absorption depends on the fibrous material inside the structure”

The authors should rewrite this paragraph. In the same paragraph, it is said that “non-fibrous materials are attracting much attention as sound absorbing materials” but perforated panels based on fibrous materials are used as example. The idea wanted to be transmitted to the reader should be expressed in a clearer way.

From line 71 to line 86: Micro perforated panels are introduced in the text in an adequate way. References from [9] to [13] are considered right. However, the scope of the micro perforated panels’ analysis could be extended with other references like:

Carbajo, J.; Ramis, J.; Godinho, L.; Amado-Mendes, P.; Alba, J. A Finite Element Model of Perforated Panel Absorbers Including Viscothermal Effects. Appl. Acoust. April 2015, 90, 1–8. https://doi.org/10.1016/j.apacoust.2014.10.013.

In the Results and discussion section: the results of the 3D model and the comparison with the 2D model and experimental results are presented in a clear way by figures 5 and 6.

In the reference [18] of the text, images of the experimental measures with the circular and squared devices are presented. The reader can search that reference in order to better understand your work, but it is advisable to include some of these images in the current article.

On the other hand, a comparison between the 3D model results with values obtained from glass wool is missing. The authors focus the first part of the introduction in presenting the distribution of cylinder and squares as a sustainable alternative to glass wool. However, some reference or some theoretical or experimental result that demonstrate this is missing.

References: please check reference [20] since it is not cited in the text.

Author Response

Please find attached the docx file.

Reviewer 2 Report

The topic of the paper “Predicted absorption performance of cylindrical and rectangular permeable membrane space sound absorbers using the three-dimensional boundary element method” is very interesting. I only have a few minor points I would like to ask the authors to address.

Section 1- In the Introduction, the first three paragraphs make some claims about the need for improved absorption in large spaces and the disadvantages of using fibrous materials, compared to the methods proposed by the authors. While agree with those statements, I think it would be appropriate to support it with some references.

Section 2- I do not have any particular comments on the analytical formulation of the problem, which looks appropriate to me.

Section 3- I can see the increased accuracy of the 3-D approach for both CPMSA and RPMSA compared with the 2-D modelling. However, there still seems to be a slight underestimation of the predicted values (3-D) compared to the measured ones at lower frequencies (typically below 250 Hz, judging from Figs. 5 and 6) for almost all cases. Could you please elaborate a bit more on these aspects?

Section 4- In the conclusions, it could be useful to have some more discussion about the benefits of the 3-D over the 2-D method when looking at other factors as well (maybe increased computational load for the model? Is this negligible?). Also, some thoughts about the “scalability” of this: apparently the tested absorbers were relatively small; what happens (both in terms of accuracy and computational load) when simulating much larger sizes of absorbers?

I think the manuscript is well-prepared and it was an interesting reading.

Author Response

Please find attached the docx file.

Reviewer 3 Report

The paper propose a numerical prediction method using the three-dimensional boundary element  method for the sound absorption performance of CPMSA and RPMSA.

The energy dissipation ratios calculated by two- and three-dimensional boundary element methods are compared with the measured absorption coefficients obtained by the reverberation room method.

The paper require the modifies reported in yellow in the attached file and the symbology of the formulas writed in the paper

Author Response

Please find attached the docx file.

Round 2

Reviewer 1 Report

All comments have been considered.